# Weight illusions explained by efficient coding based on correlated natural statistics
Paul M. Bays ⊠

In our everyday experience, the sizes and weights of objects we encounter are strongly correlated. When objects are lifted, visual information about size can be combined with haptic feedback about weight, and a naive application of Bayes' rule predicts that the perceived weight of larger objects should be exaggerated and smaller objects underestimated. Instead, it is the smaller of two objects of equal weight that is perceived as heavier, a phenomenon termed the Size-Weight Illusion (SWI). Here we provide a normative explanation of the SWI based on principles of efficient coding, which dictate that stimulus properties should be encoded with a fidelity that depends on how frequently those properties are encountered in the environment. We show that the precision with which human observers estimate object weight varies as a function of both mass and volume in a manner consistent with the estimated joint distribution of those properties among everyday objects. We further show that participants' seemingly "anti-Bayesian" biases (the SWI) are quantitatively predicted by Bayesian estimation when taking into account the gradient of discriminability induced by efficient encoding. The related Material-Weight Illusion (MWI) can also be accounted for on these principles, with surface material providing a visual cue that changes expectations about object density. The efficient coding model is further compatible with a wide range of previous observations, including the adaptability of weight illusions and properties of "non-illusory" objects. The framework is general and predicts perceptual biases and variability in any sensory properties that are correlated in the natural environment.

The size-weight illusion (SWI), one of the strongest and most robust perceptual illusions, has been studied for over a century[1]. It is commonly viewed as a challenge for computational theories that describe perception in terms of optimal probabilistic inference[2–5], because Bayes' rule is often interpreted as a bias in sensory estimation towards expectations set by prior experience. Larger objects are, on average, heavier than smaller objects in our everyday experience, yet the SWI consists of a bias in the opposite direction to this expectation: observers perceive a larger object to be lighter than a smaller object of the same weight. Previous attempts to account for the illusion have therefore appealed to violations of expectation[6], categorical representations of relative density[7], and optimality for long-distance throwing[8], among others. However, these accounts have typically relied on ad hoc assumptions about underlying mechanisms or objectives. Here we show that efficient coding based on the empirical correlation between objects' masses and volumes in the everyday environment provides a parsimonious explanation for the illusion based on normative principles. Unlike previous accounts,

this model predicts a specific relationship between bias and variability in the estimation of weight, which we confirm with quantitative fitting of empirical weight estimates generated by human observers for objects of varying size.

According to the efficient coding hypothesis, sensory systems are optimized to transmit information about the natural environment[9,10]. This can be achieved by distributing neural resources underlying encoding of sensory properties according to the relative frequency with which those properties are encountered in the world[11]. This principle has previously been invoked to explain anisotropies in human judgments about visual orientation[12,13]. Specifically, human observers are better able to discriminate small differences in angle for edges that are aligned nearly horizontally or vertically (0° or 90°, the cardinal angles) than for edges that are oriented diagonally (45° or 135°, the oblique angles): this is known as the *oblique effect*[14]. According to the efficient coding account, encoding fidelity is prioritized for cardinal orientations over obliques because cardinals are more prevalent in the environment.

University of Cambridge, Department of Psychology, Cambridge, CB2 3EB, UK. ⊠e-mail: pmb20@cam.ac.uk

In addition to the classical oblique effect, human judgments of orientation also display systematic biases, typically characterized as repulsion of an estimated angle away from the nearest cardinal axis. The fact that discriminability varies over the space of possible angles of a stimulus has been proposed as the basis of this bias. According to this account, the gradient of discriminability (the oblique effect) makes uncertainty about the angle increase with proximity to the obliques, and this shifts optimal estimates of a stimulus' orientation away from the cardinals, a phenomenon termed likelihood repulsion[13,15].

Subsequent work has generalized and refined the conditions under which this result holds[16–19] and shown that the predicted linear relationship between bias and the gradient of squared discrimination threshold, $b(x) \propto (D(x)^2)'$, is replicated across a wide range of stimulus variables[15]. However, while the theoretical model has been extended to multidimensional stimuli[20,21], empirical examples of these phenomena have so far been limited to single sensory variables. In this study, we analyzed a set of properties whose occurrence is correlated in the natural environment: the size, weight, and surface material of liftable objects.

## Methods
### Estimating natural statistics
We estimated the environmental joint distribution of object mass and volume by collating datasets 1–4 of Peters et al.[22], consisting of measured properties of a sampling of everyday objects. We excluded unliftable objects (dataset 5) on the basis that the mass of such objects would not be expected to contribute to prior expectation for lifted objects, the basis of the SWI according to an efficient coding model.

### Efficient coding model of the SWI
Following the empirical observations, our model of the SWI assumed a bivariate normal joint prior distribution over log-mass ($m$) and log-volume ($v$),

$$\begin{bmatrix} m \\ v \end{bmatrix} \sim \mathcal{N}_2 \left( \begin{bmatrix} \mu_m \\ \mu_v \end{bmatrix}, \begin{bmatrix} \sigma_m^2 & \sigma_{mv} \\ \sigma_{mv} & \sigma_v^2 \end{bmatrix} \right). \tag{1}$$

The prior distribution of log-mass conditioned on log-volume is then given by

$$m|v \sim \mathcal{N} \left( \mu_m + \frac{\sigma_{mv}}{\sigma_v^2}(v - \mu_v), \sigma_m^2 - \frac{\sigma_{mv}^2}{\sigma_v^2} \right). \tag{2}$$

Following the formulation of Wei & Stocker[15], an efficient coding of log-mass of an object with respect to its log-volume will result in a discrimination threshold or just-noticeable difference for log-mass,

$$D_m(m, v) \propto \frac{1}{p(m|v)}, \tag{3}$$

and a Bayesian estimate of log-mass based on this encoding will have bias,

$$b_m(m, v) \propto \left( D_m(m, v)^2 \right)', \tag{4}$$

with this and all derivatives taken with respect to $m$. The sign of the proportionality is positive (i.e., repulsive) for the posterior mean estimate (minimizing squared error) and all estimates that minimize loss functions with exponent ≥ 1. The s.d. of the Bayesian estimate is

$$\hat{\sigma}_m(m, v) \propto D_m(m, v)(1 + b_m'(m, v)). \tag{5}$$

We fit this model to data from the SWI study of Peters et al.[7]. Stimuli were twelve cubes comprising three different masses and four different volumes. Thirty healthy young participants (see source study for demographic details) lifted objects in pairs of the same mass but different volumes, and reported an estimate of the ratio of their masses. We assumed that the

estimated log-mass $\hat{m}_{ij}$ of an object with log-mass $m_i$ and log-volume $v_j$ is normally distributed with bias $b_{ij}$ and s.d. $\hat{\sigma}_{ij}$ given by Eqs. (4) & (5) above,

$$\hat{m}_{ij} \sim \mathcal{N} \left( m_i + b_{ij}, \hat{\sigma}_{ij}^2 \right), \tag{6}$$

in which case the log of the estimated mass ratio of two objects with the same log-mass $m_i$ and log-volumes $v_j$ and $v_k$ is then also normal with

$$\log \hat{r}_{ijk} = \hat{m}_{ij} - \hat{m}_{ik} \sim \mathcal{N} \left( b_{ij} - b_{ik}, \hat{\sigma}_{ij}^2 + \hat{\sigma}_{ik}^2 \right). \tag{7}$$

Note that the assumption of normality in log-mass estimates (and hence log mass ratio estimates) is made primarily for model simplicity and computational efficiency. A more detailed implementation of the encoding-decoding process, like that in Wei & Stocker[13], would also make predictions for higher moments of the estimate distribution, including skewness. However, these predictions would vary with model specifics, including the internal noise distribution and the loss function, whereas the relationships we rely on above (Eqs. (3)–(5)) are more general[16] and have been empirically validated for a range of stimuli[15].

**Implementing the model**. The model as described above is over-parameterized (i.e., different combinations of parameters make identical predictions), so we re-write the prior density conditional on volume (Eq. (2)) as

$$p(m|v) = \phi(m; \beta v + c_m, s) \tag{8}$$

with $\phi(x; \mu, \sigma)$ the normal p.d.f. evaluated at $x$. The new parameters with respect to those of Eq. (2) are

$$\beta = \frac{\sigma_{mv}}{\sigma_v^2}, \tag{9}$$

$$c_m = \mu_m - \frac{\sigma_{mv}}{\sigma_v^2} \mu_v = \mu_m - \beta \mu_v, \tag{10}$$

$$s^2 = \sigma_m^2 - \frac{\sigma_{mv}^2}{\sigma_v^2}. \tag{11}$$

Defining $c_b$ and $c_\sigma$ as proportionality constants in Eqs. (4) & (5), we have bias

$$b_m(m, v) = c_b(p(m|v)^{-2})' \tag{12}$$

and s.d.

$$\begin{aligned} \hat{\sigma}_m(m, v) &= c_\sigma \left( 1 + b_m'(m, v) \right) p(m|v)^{-1} \\ &= c_\sigma (1 + c_b(p(m|v)^{-2})'') p(m|v)^{-1} \end{aligned} \tag{13}$$

We used non-linear optimization to obtain maximum likelihood values for each participant of parameters $\beta$, $c_m$, and $s$, the slope, intercept, and s.d. describing the prior distribution, and $c_b$ and $c_\sigma$, the constants of proportionality for bias and s.d., respectively. Specifically, for each participant, we used a custom-coded pattern search algorithm to iteratively search for parameters that minimized the summed negative log-likelihoods of the reported mass ratio estimates on each trial, based on the mass and volumes of each pair of lifted items. To protect against local minima, each search was repeated with 100 sets of randomized starting parameter values, and the parameters corresponding to the global maximum likelihood selected. To enhance search efficiency, the fitting algorithm used the parameterization $\{\beta, c_m, s^2, \log_{10}(c_m), \log_{10}(c_\sigma)\}$ and the obtained maximum likelihood values were subsequently transformed to values of $\{\beta, c_m, s, c_m, c_\sigma\}$ for ease of interpretation. Medians calculated from

maximum likelihood fitted parameter values with $\mu_v = 6$, $\sigma_{mv} = 5$, were used as parameters of an illustrative bivariate normal prior.

### Efficient coding model of the MWI

We extend Eq. (2) in our model of the SWI to describe the prior distribution of log-mass conditioned on both log-volume and surface material as

$$m|v, M \sim \mathcal{N}(v + \overline{d}(M), s^2), \tag{14}$$

where $\overline{d}(M)$ is the mean log-density of material $M$.

 We fit data from the study of Buckingham et al.[23], which compared the perceived weights of lifted objects that appeared to be made of different materials. Stimuli consisted of three cubes, each of a different surface material: aluminum (of density 2.7 g cm$^{-3}$), wood (0.7 g cm$^{-3}$), and expanded polystyrene (0.1 g cm$^{-3}$). All three cubes were identical in mass (700 g) and volume (1000 cm$^3$), having the density of wood (0.7 g cm$^{-3}$). Twenty-five healthy young participants (see source study for demographic details) lifted each cube 15 times in a randomized sequence of triplets, reporting an absolute magnitude estimate of object weight after each lift. Each participant's magnitude estimates were z-scored in the original study (scaled and shifted to have a pooled mean of zero and s.d. of 1). The authors noted a consistent tendency for magnitude estimates to increase over the course of the experiment, so we additionally detrended the data by subtracting the mean rating from each triplet of lifts in the sequence, before calculating the s.d. of estimates for each object.

 For modeling purposes, we assume that the magnitude estimates, $\psi$, are linearly related to the perceived log-mass on each lift, i.e., $\psi_i = \gamma \hat{m}_i + c$, with $\hat{m}_i \sim \mathcal{N}(m + b_i, \hat{\sigma}_i^2)$ as above (Eq. (6)), and bias $b_i$ and variability $\hat{\sigma}_i$ for each material obtained from the corresponding prior (Eq. (14)) as in Eqs. (4) & (5). We used the same non-linear optimization method as for the SWI to find maximum likelihood parameters $s$, the s.d. of the conditional prior density; $c_b$ and $c_\sigma$, constants of proportionality for bias and s.d., respectively.

 Inferential statistics were assessed with Bayesian ANOVA in JASP[24] using the default Jeffreys-Zellner-Siow prior on effect sizes.

 The present study was not preregistered. As secondary analysis of fully anonymized data this study was exempted from ethical review, following guidelines of the University of Cambridge.

### Reporting summary

Further information on research design is available in the Nature Portfolio Reporting Summary linked to this article.

## Results

### Correlated natural statistics

We began by considering the natural statistics of size and weight. Based on a large sample of liftable objects found in everyday environments[22], we observed that mass and volume have a joint density that is approximately bivariate log-normal with a strong positive correlation (bivariate normal on log axes, Fig. 1A; correlation, $\rho = 0.67$). As a consequence, visual evidence that a lifted object has a larger or smaller volume would imply a conditional probability density for the object's mass that is log-normal with a peak at higher or lower mass, respectively. Efficient coding theory dictates that encoding resources should be allocated according to this natural frequency, inducing a gradient of discriminability for mass that reflects its conditional probability given the evidence for volume (blue curves in Fig. 1B, top and bottom).

 For two objects with different volumes (one large, one small; Fig. 1B, top and bottom) that invoke the same haptic feedback about mass when lifted (corresponding to arrow and dashed line), the differing gradients of discriminability result in likelihood functions for mass that are skewed in opposite directions. In other words, the internal representation is compatible with a range of masses that deviates further in the direction of decreasing prior probability, because those masses are encoded with less precision. This produces a relative bias in point estimates of mass (e.g., the mean posterior estimate) such that the larger object is perceived as lighter

than the small object, qualitatively matching the SWI. Figure 1C illustrates how variability (circle diameter) and bias amplitude and direction (arrows) in mass estimates are expected to vary for objects of different mass and volume, assuming a correlated joint distribution (blue contours).

### Fits to empirical data

Figure 2 shows variability (black circles) and relative bias (black arrows) estimated from empirical observations of reported weight ratios of differently sized objects lifted by human participants[7]. These results, which demonstrate a strong SWI, conform to the predicted pattern of Fig. 1C. Both estimation bias and variability depend on the combination of mass and volume in a manner qualitatively consistent with jointly correlated statistics (extreme evidence for effects of mass on bias, $BF_{incl} = 7.42 \times 10^8$, and SD, $BF_{incl} = 1.21 \times 10^9$; effects of volume on bias, $BF_{incl} = \infty$, and SD, $BF_{incl} = 3.89 \times 10^{10}$; and mass-volume interaction effects on bias, $BF_{incl} = 2.56 \times 10^5$, and SD, $BF_{incl} = 7.38 \times 10^8$).

 Red circles and arrows show mean predicted variability and bias, respectively, for the fitted efficient coding model (maximum likelihood parameters, median values: prior intercept, $c_m = 2.40$; prior slope, $\beta = 0.306$; prior s.d., $s = 3.22$; bias constant, $c_b = 0.0123$; s.d. constant, $c_\sigma = 0.0102$; individual participant parameters shown in Fig. S1A; equivalent parameters for empirical sample shown in Fig. 1A for comparison: $c_m = 1.10$, $\beta = 0.483$, $s = 0.361$).

 Figure 3A (black circles) plots the mean estimated weight ratios for individual pairs of lifted objects from which the empirical estimates in Fig. 2 are calculated, along with predictions of the fitted efficient coding model (red lines). Despite each lifted pair having equal mass, the object of smaller volume was, on average, estimated as heavier (weight ratio > 1) for every pair of objects in each mass condition, demonstrating the SWI. The strength of the SWI decreased as the ratio of the volumes decreased (left to right within each panel), and increased on average as mass of the objects increased (left-hand to right-hand panels). Each of these patterns was quantitatively reproduced by the efficient coding model (red lines), based on the non-linear relationship between mass, volume and prior probability depicted in Fig. 2 (blue contours).

 Notably, the SWI differed in strength for pairs of objects with the same volume ratio (labeled with an equals sign in Fig. 3), in most cases being stronger for the smaller pair (generating the zig-zag pattern in Fig. 3A middle and right-hand panels), and this pattern was also captured successfully by the efficient coding model. This variation arises because the model-predicted relative bias between two objects depends only indirectly on their relative volumes, and is instead mediated by the probability of the objects' masses and volumes under the prior. Thus, two objects with the same volume ratio (e.g., volume pairs A/B and C/D in Fig. 2, separated by the same distance on this logarithmic volume axis) may be situated asymmetrically relative to the prior density (contour lines) and so differ in their individual and relative biases.

 Figure 3B (black circles) plots the within-participant s.d. of weight ratio estimates in the same format as Fig. 3A. Variability in estimation increased with object mass (left-hand to right-hand panels), and for pairs with equal volume ratio, was in most cases greater for the smaller pair (zig-zag pattern in middle and right-hand panels). The efficient coding model quantitatively captured the average SD within each mass condition, the increase in SD with mass, and also the pattern of variation in SD across pairs for the heaviest objects (550 g), although it underestimated the amplitude of variation in SD across object pairs for the 150 g (and to a lesser extent the 350 g) objects.

### Variant encoding models

In addition to the encoding model based on conditional prior density, described above, we considered two alternative encoding schemes that also achieve the efficient coding objective. These variant models, presented in detail in Supplementary Text, led to qualitatively (Fig. S2) and quantitatively similar predictions (Fig. S3) to the conditional density model, also consistent with the SWI. Formal model comparison using AIC indicated that a model

based on factorizing the prior distribution into independent components fit less well, in most cases, than a model based on allocating resources according to the joint prior density, which performed very similarly to the conditional density model. The best-fitting model was the conditional density model for 53% of participants (16/30), the joint density model for 37% (11/30), and the factorized density model for 10% (3/30).

## Material-weight illusion

We next considered whether the efficient coding principle could explain the material-weight illusion (MWI)[25], in which an object with the visual appearance of dense material is perceived on lifting as lighter than a matched object of seemingly less dense material. We assumed that the visual appearance of different surface materials (Fig. 4A; objects used in ref. 23)

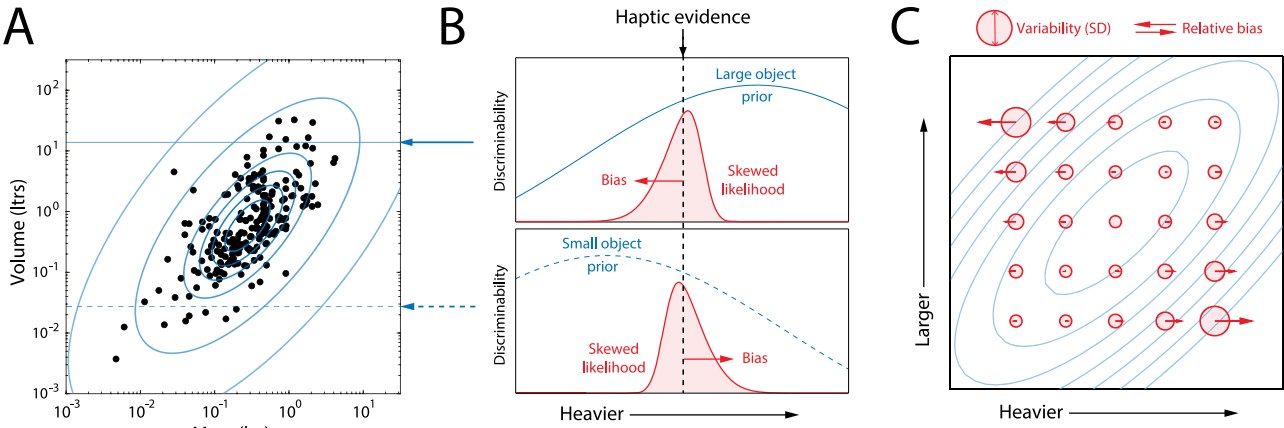

**Fig. 1 | Predictions for bias and variability based on correlated environmental statistics. A** Mass and volume of a large sample of liftable everyday objects[22] plotted on logarithmic axes. Blue contours show the best-fitting bivariate log-normal distribution (normal on these axes). **B** Illustrates estimation of mass for objects of larger (top) and smaller (bottom) volume. Blue arrows indicate corresponding volumes in (**A**). Discriminability (inverse of discrimination threshold or JND) varies in proportion to the conditional probability of object mass, given visual evidence of object volume (blue curves). For the same haptic feedback of object mass (black dashed line), the different gradients of discriminability cause likelihood functions (red) to skew in opposite directions. Red arrows show the relative directions of bias in mean posterior estimates of mass. **C** Predicted SD (circle diameters) and bias (arrows) in estimates of mass for objects with a range of true volumes and masses. Estimated weights of larger objects are underestimated relative to smaller objects of the same weight, as in the SWI.

**Fig. 2 | Summary data and fits.** Black arrows and circles plot mean relative bias (arrows) and within-participant SD (circle diameters) of human observers (*n* = 30) in estimating mass, based on reported weight ratios in a task comparing pairs of lifted objects of equal weight and different volume[7]. Red circles and arrows show corresponding mean predictions for the fitted efficient coding model with log-normal prior distribution (blue contours plot the prior density based on median fitted parameters). Row labels A–D indicate objects of the four different tested volumes (illustrated in Fig. 3C).

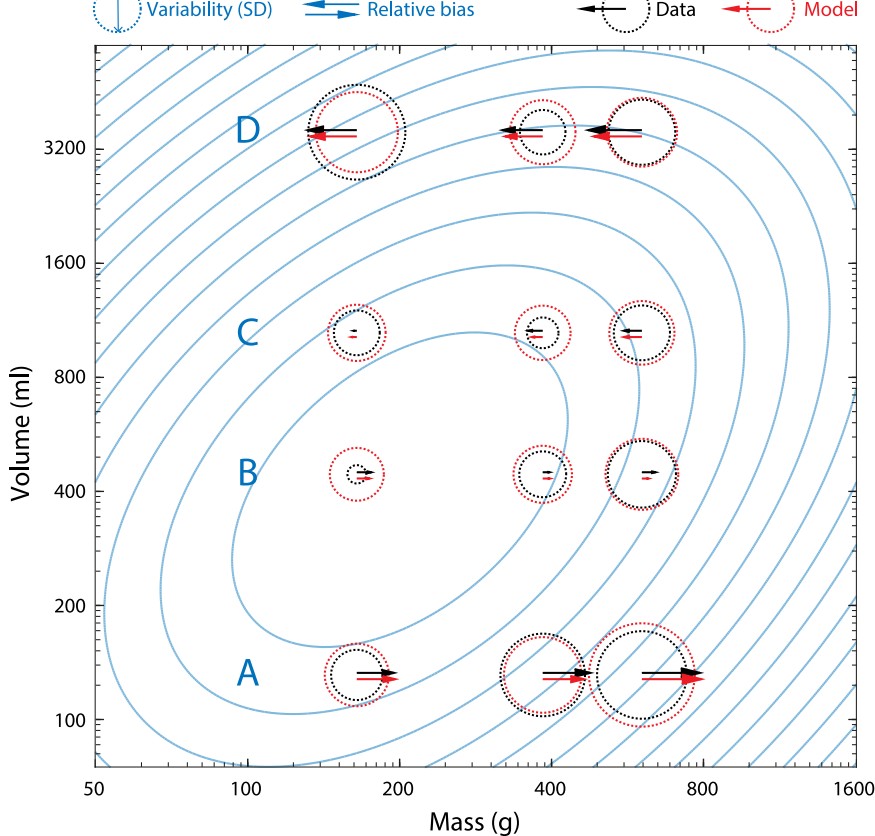

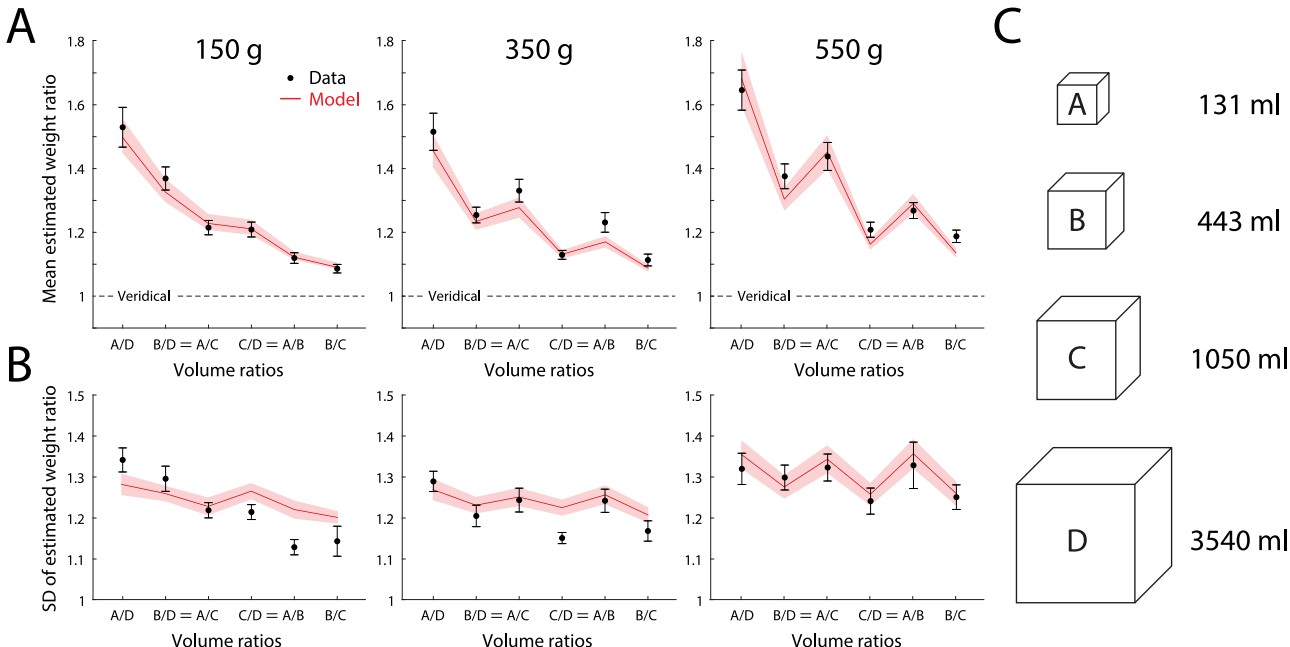

**Fig. 3 | Estimated weight ratios and model fits. A** Mean weight ratio reported by participants (black; *n* = 30) for each pairing of lifted objects, with model predictions (red). Every pair of lifted objects had equal mass; estimated weight ratios greater than one indicate illusory overestimation of mass of the smaller object (the SWI). Comparisons within each panel are plotted in decreasing order of volume ratio (largest volume discrepancy on the left; equals signs (=) indicate different object pairs with equal volume ratio. Different panels show results for three sets of objects with different common masses. Error bars and shading indicate ± 1 SEM. **B** Within-participant SD of weight ratios, plotted as in (**A**). **C** Shape and volume of the objects.

would invoke different prior distributions based on previous experience of objects made of that material, as illustrated in Fig. 4B. Specifically, we assumed the induced prior would reflect a mean density matching that of the surface material (dashed lines indicate combinations of mass and volume consistent with the density of each material).

As illustrated in Fig. 4C, the consequence of efficiently encoding object mass conditioned on its volume and surface material depends on the difference between the density of the surface material and the density of the object. If the surface material is of higher density than the object as a whole (top panel), the prior distribution conditioned on volume will peak at a higher mass than the one indicated by haptic evidence from lifting the object (vertical dashed line), resulting in a gradient of discriminability (green line) that biases the mean posterior estimate towards lower masses (red arrow). The converse holds for a surface material of lower density than the object (bottom panel). For a surface material that matches the object density (middle panel), the prior probability, and hence the discriminability, peaks at the true mass of the object, so no bias is induced. Additionally, the discriminability of mass in this case is predicted to be greater than when the surface material and object density are in conflict.

Figure 5A plots relative bias in magnitude judgments of weight when lifting the objects illustrated in Fig. 4A. Consistent with the MWI, the data show a positive bias in estimated mass for less compared to more dense surface materials, despite all the objects having the same mass and volume. Figure 5B shows the corresponding within-participant SDs (maximum likelihood parameters, median values: prior s.d., $s$ = 10.3; bias constant, $c_b$ = 0.0092; s.d. constant, $c_\sigma$ = 0.0257; individual participant parameters shown in Fig. S1B).

Note that aligning prior densities with the actual densities of the experimental materials reported by Buckingham et al.[23] is a simplification for computational convenience and to limit model flexibility: in reality, we would not expect individual observers to infer the precise density of a material from its visual appearance, and this would introduce individual variability in the alignment of prior densities (along the mass axis in Fig. 4B). However, because the different experimental materials had very different densities we would not expect this to qualitatively affect the results.

## Discussion

We have shown how weight illusions arise as an indirect consequence of efficient coding based on everyday experience with objects, in which variation in the discriminability of object properties, reflecting their relative prevalence in the environment, induces estimation biases when haptic evidence for a lifted object's mass is uncertain. This account of the SWI as an adaptation of encoding to environmental statistics may also explain why prolonged exposure to objects for which size and weight were anti-correlated caused a reversal of the illusion[26]. This would match the pattern of adaptation recently demonstrated for orientation biases after exposure to stimuli drawn from a distribution that favored oblique angles[27], and implies that efficient coding is not an immutable feature of sensory processing but is based on continual learning of environmental statistics.

This is not the first study to propose that the seemingly "anti-Bayesian" bias of the SWI might be consistent with Bayesian estimation. Peters et al.[7] suggested that, when comparing two objects, A and B, the brain evaluates three categorical hypotheses about their relative densities: that they have equal density, that A is more dense than B, or that A is less dense than B. Each category is associated with a different prior distribution over relative volume and density, and the influence of these competing priors leads to a bias in the posterior estimate of relative weight in the direction of the SWI. However, the assumption that relative density is treated categorically in weight comparison seems to have no independent justification, beyond the service it performs in the model. Moreover, while the categorical model with hand-tuned parameters reproduced the magnitude and overall pattern of decreasing SWI with decreasing volume ratio, it failed to predict the empirical differences in SWI for object pairs with the same volume ratio (marked by equals signs in Fig. 3A) and also the effects of object mass on SWI (compare panels left to right in Fig. 3A). The categorical model also makes no explicit predictions for variability of estimates, and its quantitative predictions were not evaluated or compared to data. The present model reproduced all of these elements of the SWI based on the ecological goal of efficient coding and a single prior grounded in natural statistics. In particular, and distinguishing it from previous attempts to model weight illusions, the efficient coding account made novel quantitative predictions

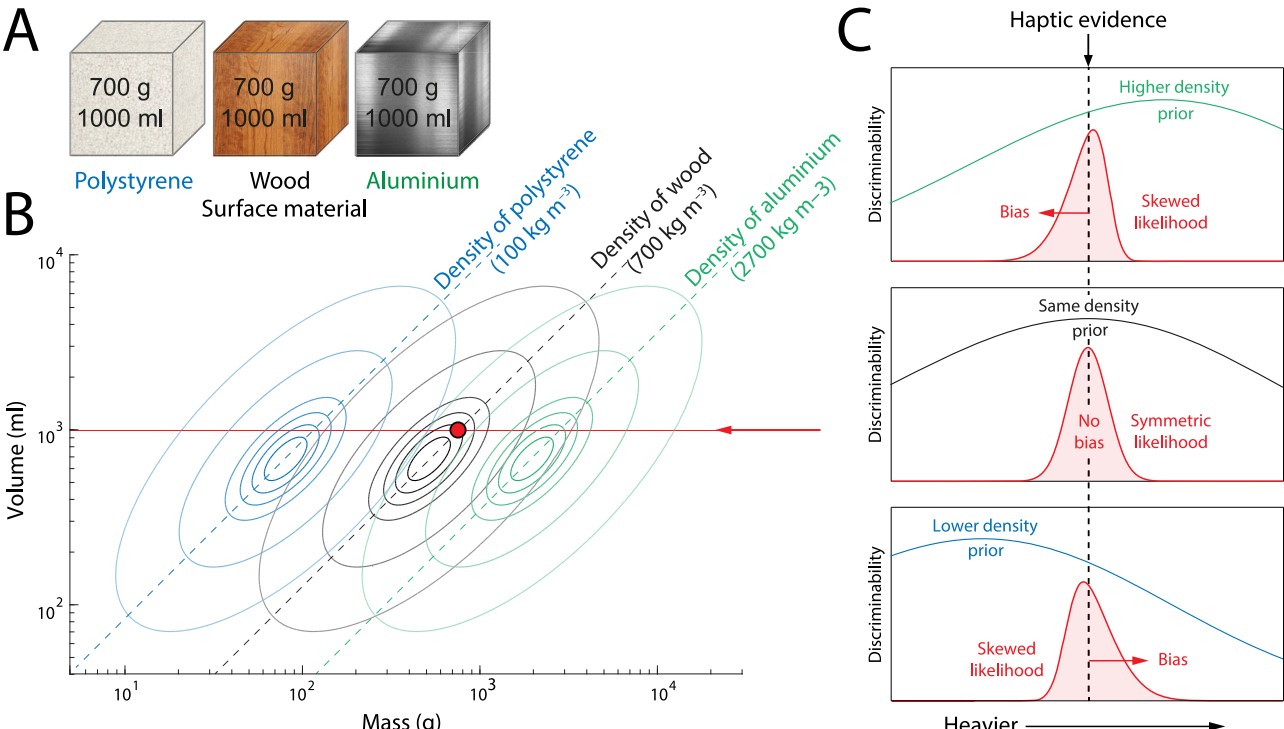

**Fig. 4 | Predicted effects of surface material on mass estimation. A** Illustration of the experimental objects, which were three cubes of identical size and weight but different surface material. **B** Contours illustrate possible prior distributions over mass and volume based on past experience of objects made from the three different materials. In the absence of haptic feedback, the expected mass-volume ratio of the three objects will approximately match the density of the surface material (dashed lines). The red disc corresponds to the actual mass and volume of all three objects.

**C** Illustrates estimation of mass for a lifted object with an expected density that is larger (top), equal (middle), or lower (bottom) than its actual density. Discriminability varies in proportion to the conditional probability of object mass, given visual evidence of object volume and object density. For the same haptic feedback (black dashed line), the different gradients of discriminability lead to different likelihood functions (red curves) and different biases (red arrows) in the mean posterior estimates.

relating bias and variability in weight estimation, which were confirmed by close fits to data from SWI and MWI experiments.

The SWI[28] and MWI[29] can be induced by preceding visual observation of an object's size, even if visual and haptic evidence of size is prevented during the lift. The magnitude of both illusions was found to be weaker in this scenario than with concurrent vision, consistent with greater uncertainty about object volume (SWI) or density (MWI), leading to a broadening of the prior distribution with respect to object mass. According to the efficient coding model, bias magnitude is proportional to the gradient (rate of change) of prior probability with respect to mass: when the prior is broader and flatter (increased s.d.), the strength of bias is expected to decrease. Similarly, Ellis & Lederman[30] observed a weaker SWI based on visual evidence of size alone than with both visual and haptic feedback, consistent with the efficient coding account. They also observed that the SWI based on haptics alone was stronger than with vision only, suggesting that haptic feedback obtained from holding an object supported a more precise estimate of the object's volume than visual observation[31,32].

The SWI persists over the repeated lifting of the same objects, but anticipatory motor responses, in the form of grip and load forces, adapt over only a few lifts to become appropriately scaled for veridical object weights[33,34]. This dissociation may reflect the different goals of perceptual and sensorimotor systems. Whereas the SWI can be interpreted as the perceptual system attempting to minimize estimation error while taking into account anisotropy in encoding fidelity, the sensorimotor system's goal is to apply the correct forces to smoothly lift each object without slipping. Lift kinematics and haptic feedback provide strong error signals about inappropriate grasp that can be used for corrective adjustment of force on the next lift of the same object, and iterated until the desired kinematics are achieved. In contrast, the SWI is a bias in relative judgments of weight that, according to the present account, reflects optimal perception given the combination of haptic and visual feedback experienced during a lift. As such, there is no error signal generated by repeated lifting that would lead this perception to change.

The present study draws on previous work that has formulated the goal of efficient coding in terms of allocation of Fisher Information[11,13,16], which can, in turn, be related to discriminability via the Cramér-Rao bound[35]. The model of weight illusions presented here, therefore, makes clear predictions for discrimination thresholds (JNDs), in addition to estimation bias and variability. However, relatively few empirical studies have measured weight discrimination performance while varying secondary object properties such as size. One such study[36] reported that weight discriminability was maximal for "non-illusory" objects, i.e., those whose apparent weight on lifting is the same whether visual evidence about the object's material and size is present or absent[6]. This aligns quite precisely with predictions of the efficient coding model of the SWI and MWI, in which biases arise from and vary in proportion to the gradient of (squared) discriminability (Methods, Eq. (4)), with the result that peak discriminability coincides with zero estimation bias. A further prediction of the model, that might be tested in future work, is that this peak corresponds to the empirical maximum of the conditional prior density, i.e., that the "non-illusory" weight for an object is the weight that is most probable given its visual properties, based on natural statistics.

## Limitations

We performed independent model fits to data from each of the thirty participants in the source study of the SWI. Examination of individual fits (Fig. S1A) indicated variability in maximum likelihood parameter estimates, with some significant outliers, although it is unclear to what extent this reflects true inter-individual differences versus variation in data quality or the relatively small number of mass ratio judgments (median 154) per participant. Most individual fits clustered around the median parameter

**Fig. 5 | Material-weight illusion data and fits.** Black circles indicate mean relative bias (**A**) and mean within-participant SD (**B**) of human observers (*n* = 25) in estimating the mass of lifted objects with different surface materials but identical mass and volume[23]. Error bars indicate ± 1 within-subject SEM. Red lines show predictions of the fitted efficient coding model.

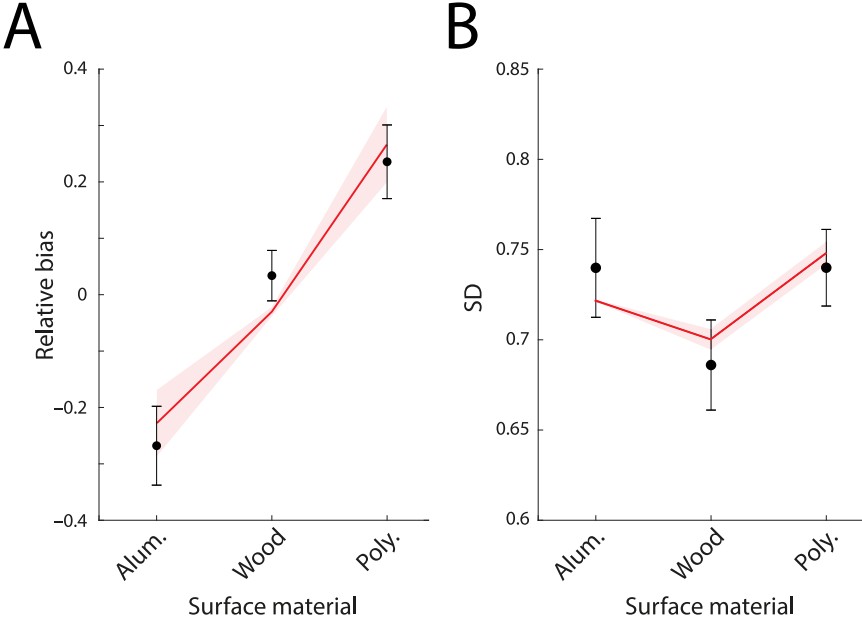

values, corresponding to prior densities for log-mass and log-volume comparable to the empirical sample of liftable objects collated by Peters et al.[22], with respect to intercept and slope. However, there was a notable discrepancy between the conditional prior s.d. parameters obtained by model fitting (median 3.22 log-units) and the equivalent measure in the empirical sample (0.361 log-units), suggesting that the internal prior distribution used by participants for estimation was substantially broader than the range of masses per volume obtained by the previous authors by measuring everyday objects.

There are a number of possible explanations for this discrepancy: it could simply reflect the inherent difficulty of obtaining a sample of object properties that is representative of human experience, or it could indicate that the prior applied by participants in the estimation task differs from our assumptions, e.g., is based on a broader category than liftable objects. An alternative interpretation for the broadness of estimated priors is that participants have relatively weak expectations about mass-volume relationships in novel objects and accordingly distribute their probability mass over a large range. One simplifying assumption of the model is that participants could visually estimate the volumes of the different objects without noise. This could contribute to the discrepancy in s.d., because uncertainty in estimating object volume would be expected to increase uncertainty in the conditional probability of mass, exceeding the natural variation in mass of objects of that precise volume. These possibilities could be addressed in future experiments that directly examine observers' expectations about mass and volume, for example, by asking them to estimate object weights based solely on visual appearance, and additionally report their uncertainty in the estimates.

Biased estimates of weight in the model ultimately arise from the fact that size and weight are correlated in our everyday experience of lifted objects. By symmetry, this principle should also predict a bias in estimating an object's size due to its weight (an inverse, "weight-size" illusion). However, we might expect this bias to be significantly weaker and so more difficult to detect than the size-weight illusion, because of the relatively high precision evidence vision provides about object size. We know of only one study that has looked for an influence of weight on size estimates: Smeets et al.[37] obtained free-magnitude estimates of the size of objects lifted at a distance via a pulley mechanism, and did not detect an influence of lifted weight on size judgments. However, it is possible that the combination of indirect lifting mechanism (participants pulled horizontally on a string) and restricted visual input (objects were 10 cm luminous balls viewed briefly at ≥2 m distance in total darkness) disrupted the association between haptic and visual feedback of the object necessary for an illusory effect. Future studies could employ different experimental methods to add to the evidence regarding the putative weight-size illusion.

The present results do not uniquely specify a coding scheme for the relevant object properties, and a number of different encodings consistent with the efficient coding goal are compatible with the observed biases and variability in estimation. In addition to the conditional density model of the SWI presented in the main text, which implicitly assumes that object volume is encoded separately from object mass, we also considered a variant that encodes the two properties jointly, by factorizing their joint prior density into two orthogonal components, as well as a variant where encoding of mass is efficient with respect to the joint, instead of conditional, prior. These alternative models were found to make qualitatively similar predictions consistent with the SWI, and fitting showed them to have similar compatibility with the data (see Supplementary Text for full details). While the predictions for bias differed only weakly within the experimental range of masses and volumes, larger deviations might be observed for more extreme pairings of object properties, potentially allowing future studies to better discriminate between alternative encoding schemes. However, the largest discrepancies between models will tend to coincide with the lowest prior probabilities (e.g., objects with unnaturally high or low density), which could prove challenging to evaluate experimentally.

The conditional density model, in combination with the bivariate Gaussian prior, admits a particularly simple implementation: encoding an object's log-weight in relative terms as a deviation from its expected log-weight given object volume. Allocating coding resources (e.g., neuronal tuning density) preferentially to smaller deviations would achieve efficiency with respect to natural statistics (note this scheme depends on properties of the bivariate Gaussian prior and might not be realizable for a different prior distribution). Early accounts of weight illusions frequently appealed to the contrast between expected (based on visual properties) and actual sensory feedback experienced during the lift as the basis of the illusion[6,38], i.e., a larger object feels lighter when lifted because it is lighter than *expected*. While this informal account is only descriptive of the illusion, the present results present a normative basis for biasing perception relative to the expected mass.

## Conclusions

We presented a computational account of weight illusions in which biased perception reflects Bayes-optimal estimation given the variation in the

fidelity with which object weights are encoded in the neural system: variation that, in turn, reflects an efficient allocation of coding resources based on the natural frequency with which objects with different properties are encountered. This account made specific predictions connecting bias and variability of weight estimates, which we confirmed with quantitative fitting of human participant data from studies of the Size-Weight and Material-Weight Illusions.

## Data availability

Data related to this study is available at https://doi.org/10.17863/CAM. 108265.

## Code availability

Code related to this study is available at https://doi.org/10.17863/CAM. 108265.

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

## Acknowledgements

We thank Máté Lengyel and Alan Stocker for helpful discussion and comments, and Megan Peters and Gavin Buckingham for sharing data. This work was funded by the Wellcome Trust (grant 106926). The funders had no role in study design, data collection and analysis, the decision to publish, or the preparation of the manuscript.

## Competing interests

The author declares no competing interests.
