## [Transparent Peer Review file · Communications Psychology]

Weight illusions explained by efficient coding based on correlated natural statistics

Corresponding Author: Professor Paul Bays

Version 0:

Decision Letter:

Dear Professor Bays,

Thank you for your patience during the peer-review process. Your manuscript titled "Weight illusions explained by efficient coding based on correlated natural statistics" has now been seen by 3 reviewers, and I include their comments at the end of this message. They find your work of interest but raised some important points. We are interested in the possibility of publishing your study in Communications Psychology, but would like to consider your responses to these concerns and assess a revised manuscript before we make a final decision on publication.

We therefore invite you to revise and resubmit your manuscript, along with a point-by-point response to the reviewers. Please highlight all changes in the manuscript text file.

Editorially, we consider it important that the revised manuscript clarifies the modeling questions raised by the reviewers. Please ensure a link to the code is available for anonymous review.

I am attaching an Editorial Requests Table that details critical reporting requirements for the revised manuscript. Please attend to each item and ensure your manuscript is fully compliant. We are requesting that your manuscript aligns with these requirements as this facilitates the evaluation of your manuscript, reducing delays in re-review and potential future acceptance. If your revised manuscript is not aligned with these requests on major issues, such as those concerning statistics, it may be returned to you for further revisions without re-review. Additional information can be found in our style and formatting guide <https://www.nature.com/documents/commspsychol-style-formatting-guide-accept.pdf>>Communications Psychology formatting guide.

Please use the following link to submit your

- revised manuscript,
- point-by-point response to the referees' comments,
- cover letter (as a separate document),
- the Editorial Policy Checklist (see below),
- the Reporting Summary (see below), and
- the completed Editorial Request Table (attached):

Link Redacted

We hope to receive your revised paper within 8 weeks; please let us know if you aren't able to submit it within this time so that we can discuss how best to proceed. If we don't hear from you, and the revision process takes significantly longer, we may close your file. In this event, we will still be happy to reconsider your paper at a later date, provided it still presents a

significant contribution to the literature at that stage.

Best regards,

Jennifer Bellingtier

Jennifer Bellingtier, PhD
Senior Editor
Communications Psychology

REVIEWER EXPERTISE:

Reviewer #1 Haptic/weight perception, sensorimotor control

Reviewer #2 Bayesian models of perception, biases in perceptual estimation

Reviewer #3 Computational models of perception, Bayes, normative modelling

REVIEWER REPORTS:

Reviewer #1 (Remarks to the Author):

I reviewed the manuscript "Weight illusions explained by efficient coding based on correlated natural statistics" by Bays submitted for publication in Communications Psychology. The author proposes an explanation of the size-weight illusion (SWI) based on the notion of efficient coding and develops a model that accounts for aspects of SWI including biases of human observers. The underlying idea is that observers should be more sensitive to changes in a given property when it falls in an expected (previously experienced) range than when it does not. When objects are far from expectations, the model predicts (in agreement with experimental observations) that estimates of those properties are relatively coarse. Overall I think the manuscript is interesting and well written. I think the manuscript could be improved by more extensive comparison of the proposed model to other explanations for SWI. The manuscript is based only on modeling of existing experimental data; I cannot evaluate how this fits with the scope of Communications Psychology.

Comments:

1. Contribution. The author writes that "[previous accounts of SWI] have typically relied on ad hoc assumptions about underlying mechanisms or objectives" (P1, L29-30) and also that "Unlike previous accounts [of SWI], this model quantitatively fits both the bias and variability of empirical weight estimates generated by human observers." (P1, L33-34). I take it that the author sees these as the major contributions of the present model. It would be helpful to come back to these terms in the discussion when the author describes how the proposed model fits with or improves on previous models, especially because the manuscript does not include an original experiment which may leave the reader a bit underwhelmed as to the significance of this model.

2. Material weight illusion. The author objects to the "categorical treatment of relative density" in Peters et al. 2016 as "not clearly justified in this account and appears to be largely ad hoc" but I think the author needs to provide more explanation of what precisely the objection is, and specifically how it is different from the proposed explanation of the material weight illusion, in which "we assumed the induced prior would reflect a mean density matching that of the surface material" (P6, L120-121) which seems quite categorical, and even a bit hard to imagine (e.g. recognizing some metallic surface as aluminium rather than steel or lead or another metal and "looking up" its specific density). It would be helpful to explain in more detail how the explanation of MWI is different from the idea of categorical hypotheses regarding densities, as the author objects to in Peters et al. 2016.

3. Owing to my bias as an experimentalist, I would still find it more convincing to test the proposed model on a new dataset or unmodeled data -- or to use it to predict some other phenomenon. Are there any features of this model that would predict (surprising?) observable behaviors? I am not asking for a new experiment, but it would be nice to see some further proposed applications.

Minor: The second paragraph of Introduction is quite technical and difficult to follow. A more plain language summary would be helpful.

Reviewer #2 (Remarks to the Author):

This paper presents a novel formal explanation for the size-weight illusion (SWI), a classical perceptual illusion, in terms of Bayesian inference constrained by efficient coding. The account is further extended to the Material-Weight Illusion.

The paper describes a generalization of the efficient coding model of Wei & Stocker 2015, applies these to data of naturalistic size&weight distributions from Peters et al 2015, and shows good fit of the resulting model to behavioral data from Peters et al 2016. In a nutshell, the SWI is attributed to efficient coding adapting to correlated statistics of size and weight in real-world objects, resolving the puzzle posed by its apparent anti-Bayesian nature.

The paper offers a compelling and original account of a long-standing perceptual illusion, and I have no doubt it will be of broad interest. The paper is rigorous and of high theoretical and analytical quality. I have read the paper very thoroughly and believe it is essentially ready for publication, with some minor questions remaining.

In evaluating the paper, I would have liked to look at the paper's code, but the DOI link given under Data availability is unfortunately not yet (I presume) active. My assessment is thus based on the paper itself only.

I have the following questions:

- The model is fitted using maximum likelihood for a log-normal response function using the biases and variances predicted by the Wei&Stocker model. Is this specific formulation grounded in the mathematical formulation of the underlying Wei&Stocker efficient coding model (a cascade of encoding and decoding), or is this for convenience?

- In lines 244-265, I presume the human data are at some point log-transformed, but I didn't understand where. Are they log-transformed already before z-scoring? Or before subtracting the sequence mean? At any rate, in 259, the magnitude estimates are assumed to be linearly linked to the perceived log-mass

Reviewer #3 (Remarks to the Author):

This is a very well-executed theoretical study that explains classic "anti-Bayesian" weight illusions (SWI and MWI) using efficient coding principles. I only have some minor comments/questions as outlined below:

1. In Fig. 1, the author simulated the prediction of an efficient coding Bayesian model, using mass-volume prior measured by Peters et al., (2015). For comparisons with empirical data in Fig. 2 and Fig. 3, the (conditional) prior was parameterized and fitted to psychophysics data.

Could the author elaborate on how the estimated prior compares to the natural prior derived from natural objects? Figures 1 and 2 are presented on different scales, so a visual comparison is a bit difficult. How well does the model perform if the natural prior from Fig. 1 is used without any fitting procedures (aside from maybe adjustments to the overall noise level in the likelihood)? Additionally, are there significant variations in the estimated prior across individual subjects?

2. Please include the details of the MLE procedure (line 238 onwards). Was it based on the predicted distribution of the perceived weight ratio at the level of individual trials?

3. Typo in line 91: "... the object of larger volume was on average estimated as heavier ...". Should be "smaller volume".

4. The modeling assumption here (especially for the material-weight illusion) would suggest that haptic encoding of weight/force is dynamically adjusted based on visual (size, material) cues. If so, one would expect weight discrimination threshold profile changes depending on visual contexts. This is outside the scope of this study, but I was wondering if the author could comment a bit more on this (e.g., if there's any empirical evidence along this line in the literature).

5. I found the additional models with factorized and joint efficient coding on weight and volume to be particularly interesting. As the author noted, there has been little prior research addressing this issue (multivariate code) empirically. However, here all three models make very similar predictions in terms of bias/variance prediction (Fig. S2). The author might consider highlighting this discussion more in the main text.

EDITORIAL POLICIES

We ask that you ensure your manuscript complies with our editorial policies and reporting requirements.

To that end, we require revised manuscripts to be accompanied by two completed items: a reporting summary that collects information on study design and procedure, and an editorial policy checklist that verifies compliance with all required editorial policies.

- <https://www.nature.com/documents/nr-reporting-summary.zip>>Nature Research Reporting Summary

- <https://www.nature.com/documents/nr-editorial-policy-checklist.pdf>>Editorial Policy Checklist

All points on the policy checklist must be addressed. Your revised manuscript can only be sent back to the referees if these checklists are completed and uploaded with the revision.

Notes: If you have submitted a Stage 1 Registered Report, Review, Primer, Comment, or Perspective you do not need to submit these forms. If you have already submitted these forms, you may disregard this request.

** Visit Nature Research's author and referees' website at <http://www.nature.com/authors>>www.nature.com/authors for information about policies, services and author benefits**

If you experience problems in linking your ORCID, please contact the <http://platformsupport.nature.com/>>Platform Support Helpdesk.

Version 1:

Decision Letter:

Dear Professor Bays,

Your manuscript titled "Weight illusions explained by efficient coding based on correlated natural statistics" has now been seen by our reviewers, whose comments appear below. In light of their advice I am delighted to say that we are happy, in principle, to publish a suitably revised version in Communications Psychology.

We therefore invite you to revise your paper one last time to address the remaining concerns of our reviewers and a list of editorial requests. At the same time we ask that you edit your manuscript to comply with our format requirements and to maximise the accessibility and therefore the impact of your work.

EDITORIAL REQUESTS:

SUBMISSION INFORMATION:

In order to accept your paper, we require the files listed at the end of the Editorial Requests Table; the list of required files is also available at <https://www.nature.com/documents/commsj-file-checklist.pdf> .

OPEN ACCESS:

Communications Psychology is a fully open access journal. Articles are made freely accessible on publication. For further information about article processing charges, open access funding, and advice and support from Nature Research, please

visit <https://www.nature.com/commpsychol/open-access>

* **DATA AVAILABILITY:**

Link Redacted

Best regards,

Jennifer Bellingtier

Jennifer Bellingtier, PhD
Senior Editor
Communications Psychology

REVIEWERS' EXPERTISE:

Reviewer #1 Haptic/weight perception, sensorimotor control
Reviewer #2 Bayesian models of perception, biases in perceptual estimation
Reviewer #3 Computational models of perception, Bayes, normative modelling

REVIEWERS' COMMENTS:

Reviewer #1 (Remarks to the Author):

I appreciate the author's detailed responses to my comments and I am satisfied with the revised manuscript.

Reviewer #2 (Remarks to the Author):

The author has successfully addressed the questions raised in my initial review.

I also appreciate the author making the code repository available. I have studied it and believe it to be in good shape.

I believe the paper is ready for publication.

Reviewer #3 (Remarks to the Author):

I want to thank the author for the thorough revision and reply. All of my previous questions have been fully addressed.

We would like to thank all the reviewers for their thoughtful and constructive comments on our manuscript, which we have addressed with substantially revised text, new analyses and figures. We respond point-by-point below.

Reviewer #1 (Remarks to the Author):

I reviewed the manuscript “Weight illusions explained by efficient coding based on correlated natural statistics” by Bays submitted for publication in Communications Psychology. The author proposes an explanation of the size-weight illusion (SWI) based on the notion of efficient coding and develops a model that accounts for aspects of SWI including biases of human observers. The underlying idea is that observers should be more sensitive to changes in a given property when it falls in an expected (previously experienced) range than when it does not. When objects are far from expectations, the model predicts (in agreement with experimental observations) that estimates of those properties are relatively coarse. Overall I think the manuscript is interesting and well written. I think the manuscript could be improved by more extensive comparison of the proposed model to other explanations for SWI. The manuscript is based only on modeling of existing experimental data; I cannot evaluate how this fits with the scope of Communications Psychology.

Comments:

1. Contribution. The author writes that “[previous accounts of SWI] have typically relied on ad hoc assumptions about underlying mechanisms or objectives” (P1, L29-30) and also that “Unlike previous accounts [of SWI], this model quantitatively fits both the bias and variability of empirical weight estimates generated by human observers.” (P1, L33-34). I take it that the author sees these as the major contributions of the present model. It would be helpful to come back to these terms in the discussion when the author describes how the proposed model fits with or improves on previous models, especially because the manuscript does not include an original experiment which may leave the reader a bit underwhelmed as to the significance of this model.

Thanks very much for this suggestion. We have emphasized this in a new Conclusions paragraph and also added the following to the Discussion:

“...The present model reproduced all of these elements of the SWI based on the ecological goal of efficient coding and a single prior grounded in natural statistics. **In particular, and distinguishing it from previous attempts to model weight illusions, the efficient coding account made novel quantitative predictions relating bias and variability in weight estimation, which were confirmed with fits of unprecedented accuracy to data from SWI and MWI experiments.**”

2. Material weight illusion. The author objects to the “categorical treatment of relative density” in Peters et al. 2016 as “not clearly justified in this account and appears to be largely ad hoc” but I think the author needs to provide more explanation of what precisely the objection is, and specifically how it is different from the proposed explanation of the material weight illusion, in which “we assumed the induced prior would reflect a mean density matching that of the surface material” (P6, L120-121) which seems quite categorical, and even a bit hard to imagine (e.g. recognizing some metallic surface as aluminium rather than steel or lead or another metal and “looking up” its specific density). It would be helpful to explain in more detail how the explanation of MWI is different from the idea of categorical hypotheses regarding densities, as the author objects to in Peters et al. 2016.

The SWI model of Peters et al. is based on an assumption that, when comparing the weight of two objects, the brain evaluates three categorical hypotheses about the object pair: equal density, A more dense than B, B more dense than A. Note that the categorization is regarding the *relative* density of two objects, not a property of an individual object: as a result the model applies only to the situation where two objects are compared.

The reviewer suggests that our model of the MWI involves a similar categorization, which in this case would mean assigning an individual object to a particular category of material (e.g. polystyrene) based on visual appearance, and then selecting a prior based on that category. However, while it is certainly possible the brain explicitly distinguishes different categories of material, our model does not actually make or rely on this assumption. The only required assumptions are that prior expectations about an object’s mass-volume relationship are based on past experience of objects with similar visual properties (colour, texture, reflectance, etc), and that most objects one encounters with visual properties matching, e.g., the polystyrene-covered cube in Buckingham et al. (2009) actually are made of polystyrene and thus have densities that are typical of polystyrene.

For modelling purposes, we assume the mass-volume prior is aligned exactly with the true density of the surface material used in the study, although with a bivariate Gaussian distribution intended to capture an observer’s uncertainty over density. In practice, the prior elicited by the visual appearance of an object in an individual observer is likely to be only roughly aligned with the true density of its surface material. However, the MWI is typically observed when comparing materials with very different densities, e.g. metal vs wood, which we would expect to elicit distinct expectations in most observers, and our qualitative account of the phenomenon only requires that observers’ internal estimates of object density roughly scale with the density of its surface material.

Based on the reviewer's comment, we have clarified some of the description of the Peters et al. model and added the following to the text in relation to our model of the MWI:

“Note that aligning prior densities with the actual densities of the experimental materials reported by Buckingham et al. (2009) is a simplification for computational convenience and to constrain model flexibility: in reality we would not expect individual observers to infer the precise density of a material from its visual appearance, and this would introduce individual variability in the alignment of prior densities (along the mass axis in Fig. 4B). However, because the different experimental materials had very different densities we would not expect this to qualitatively affect the results.”

3. Owing to my bias as an experimentalist, I would still find it more convincing to test the proposed model on a new dataset or unmodeled data -- or to use it to predict some other phenomenon. Are there any features of this model that would predict (surprising?) observable behaviors? I am not asking for a new experiment, but it would be nice to see some further proposed applications.

The size-weight illusion has been a focus of research for so long, it isn't trivial to come up with genuinely new experiments. However, we have now added several suggestions for future research to the Discussion, including the following:

*“...One such study (Ross & Gregory, 1970) reported that weight discriminability was maximal for “non-illusory” objects, i.e., those whose apparent weight on lifting is the same whether visual evidence about the object's material and size is present or absent. This aligns quite precisely with predictions of the efficient coding model of the SWI and MWI, in which biases arise from and vary in proportion to the gradient of (squared) discriminability (Methods, Eq. 4), with the result that peak discriminability coincides with zero estimation bias. **A further prediction of the model, that might be tested in future work, is that this peak corresponds to the empirical maximum of the conditional prior density, i.e., that the “non-illusory” weight for an object is the weight that is most probable given its visual properties, based on natural statistics.**”*

*“...One simplifying assumption of the model is that participants could visually estimate the volumes of the different objects without noise. This could contribute to the discrepancy in s.d., because uncertainty in estimating object volume would be expected to increase uncertainty in the conditional probability of mass, exceeding the natural variation in mass of objects of that precise volume. **These possibilities could be addressed in future experiments that directly examine observers' expectations about mass and volume, for example, by asking them to estimate object weights***

based solely on visual appearance, and additionally report their uncertainty in the estimates.”

Minor: The second paragraph of Introduction is quite technical and difficult to follow. A more plain language summary would be helpful.

We have re-written and expanded this section to make it more accessible, including writing out the meaning of the formula in plain language:

“According to the efficient coding hypothesis, sensory systems are optimized to transmit information about the natural environment (Attneave, 1954; Barlow, 1961). This can be achieved by distributing neural resources underlying encoding of sensory properties according to the relative frequency with which those properties are encountered in the world (Ganguli & Simoncelli, 2014). This principle has previously been invoked to explain anisotropies in human judgments about visual orientation (Girshick et al., 2011; Wei & Stocker, 2015). Specifically, human observers are better able to discriminate small differences in angle for edges that are aligned nearly horizontally or vertically (0° or 90°, the cardinal angles) than for edges that are oriented diagonally (45° or 135°, the oblique angles): this is known as the oblique effect (Appelle, 1972). According to the efficient coding account, encoding fidelity is prioritized for cardinal orientations over obliques because cardinals are more prevalent in the environment.

In addition to the classical oblique effect, human judgements of orientation also display systematic biases, typically characterized as a repulsion of estimated angle away from the nearest cardinal axis. The fact that discriminability varies over the space of possible angles of a stimulus has been proposed as the basis of this bias. According to this account, the gradient of discriminability (the oblique effect) makes uncertainty about angle increase with proximity to the obliques, and this shifts optimal estimates of a stimulus’ orientation away from the cardinals, a phenomenon termed likelihood repulsion (Wei & Stocker, 2015, 2017).

Subsequent work has generalized and refined the conditions under which this result holds (Hahn & Wei, 2024; Mao & Stocker, 2024; Morais & Pillow, 2018; Prat-Carrabin & Woodford, 2021) and shown that the predicted linear relationship between bias and the gradient of squared discrimination threshold, $b(x) \propto (D(x)^2)$, is replicated across a wide range of stimulus variables (Wei & Stocker, 2017).”

Reviewer #2 (Remarks to the Author):

This paper presents a novel formal explanation for the size-weight illusion (SWI), a classical perceptual illusion, in terms of Bayesian inference constrained by efficient coding. The account is further extended to the Material-Weight Illusion.

The paper describes a generalization of the efficient coding model of Wei & Stocker 2015, applies these to data of naturalistic size&weight distributions from Peters et al 2015, and shows good fit of the resulting model to behavioral data from Peters et al 2016. In a nutshell, the SWI is attributed to efficient coding adapting to correlated statistics of size and weight in real-world objects, resolving the puzzle posed by its apparent anti-Bayesian nature.

The paper offers a compelling and original account of a long-standing perceptual illusion, and I have no doubt it will be of broad interest. The paper is rigorous and of high theoretical and analytical quality. I have read the paper very thoroughly and believe it is essentially ready for publication, with some minor questions remaining.

In evaluating the paper, I would have liked to look at the paper's code, but the DOI link given under Data availability is unfortunately not yet (I presume) active. My assessment is thus based on the paper itself only.

Apologies for this, the DOI can only be activated once we finalize the submission to the repository. The reviewer can download the code and data from:

<https://bayslab.org/public/sizeweight.zip>

<https://bayslab.org/public/sizeweight/>

I have the following questions:

- The model is fitted using maximum likelihood for a log-normal response function using the biases and variances predicted by the Wei&Stocker model. Is this specific formulation grounded in the mathematical formulation of the underlying Wei&Stocker efficient coding model (a cascade of encoding and decoding), or is this for convenience?

The bias and variance of estimated log-weights in our model matches those derived by Wei & Stocker (2017) on the basis of an encoding-decoding cascade with efficient coding and Bayesian decoding. However the assumption that log-weight is normally distributed with those moments is indeed mostly for computational convenience. The Wei & Stocker (2015) implementation also makes specific predictions for higher

moments (e.g. skewness) of the distribution, but (a) analytical formulae have not been derived for these higher moments, (b) the predictions for bias and variance have been empirically validated for many kinds of stimuli (Wei & Stocker, 2017) unlike predictions for higher moments, and (c) the relationship between bias and variance has been shown to hold more generally (e.g. Morais & Pillow, 2018), requiring fewer assumptions than the Wei & Stocker (2015) implementation.

We have added the following to Methods:

“Note that the assumption of normality in log-mass estimates is made primarily for model simplicity and computational efficiency. A more detailed implementation of the encoding-decoding process, like that in Wei & Stocker (2015), would make predictions also for higher moments of the estimate distribution, including skewness. However, these predictions would vary with model specifics, including the internal noise distribution and the loss function, whereas the relationships we rely on above (Eqs. 3–5) are more general (Morais & Pillow, 2018) and have been empirically validated for a range of stimuli (Wei & Stocker, 2017).”

- In lines 244-265, I presume the human data are at some point log-transformed, but I didn't understand where. Are they log-transformed already before z-scoring? Or before subtracting the sequence mean? At any rate, in 259, the magnitude estimates are assumed to be linearly linked to the perceived log-mass

Because we assume the magnitude estimates in the MWI data are linearly related to the estimated log-mass (q.v., Fechner's law), we don't need to log-transform them: they are already in the right form for fitting the model, which makes predictions for the bias and variance of log-mass estimates.

Reviewer #3 (Remarks to the Author):

This is a very well-executed theoretical study that explains classic “anti-Bayesian” weight illusions (SWI and MWI) using efficient coding principles. I only have some minor comments/questions as outlined below:

1. In Fig. 1, the author simulated the prediction of an efficient coding Bayesian model, using mass-volume prior measured by Peters et al., (2015). For comparisons with empirical data in Fig. 2 and Fig. 3, the (conditional) prior was parameterized and fitted to psychophysics data.

Could the author elaborate on how the estimated prior compares to the natural prior derived from natural objects? Figures 1 and 2 are presented on different scales, so a

visual comparison is a bit difficult. How well does the model perform if the natural prior from Fig. 1 is used without any fitting procedures (aside from maybe adjustments to the overall noise level in the likelihood)? Additionally, are there significant variations in the estimated prior across individual subjects?

Thanks for raising these important points. We now explicitly compare the model estimated prior parameters to those of the environmental prior plotted in Fig 1, and additionally plot individual fitted parameters of both SWI and MWI models as Supplementary Figure S1A & B. The priors are broadly compatible, except with respect to variability, where the conditional distribution of masses based on sampling everyday objects is notably less broad than the model-estimated prior density. We address these findings in the Discussion as follows:

“We performed independent model fits to data from each of the thirty participants in the source study of the SWI. Examination of individual fits (Fig. S1A) indicated variability in maximum likelihood parameter estimates, with some significant outliers, although it is unclear to what extent this reflects true inter-individual differences versus variation in data quality or the relatively small number of mass ratio judgements (median 154) per participant. Most individual fits clustered around the median parameter values, corresponding to prior densities for log-mass and log-volume comparable to the empirical sample of liftable objects collated by Peters et al. (2015), with respect to intercept and slope. However, there was a notable discrepancy between the conditional prior s.d. parameters obtained by model fitting (median 3.22 log-units) and the equivalent measure in the empirical sample (0.361 log-units), suggesting that the internal prior distribution used by participants for estimation was substantially broader than the range of masses per volume obtained by the previous authors by measuring everyday objects.

There are a number of possible explanations for this discrepancy: it could simply reflect the inherent difficulty of obtaining a sample of object properties that is representative of human experience, or it could indicate that the prior applied by participants in the estimation task differs from our assumptions, e.g. is based on a broader category than liftable objects. An alternative interpretation for the broadness of estimated priors is that participants have relatively weak expectations about mass-volume relationships in novel objects and accordingly distribute their probability mass over a large range. One simplifying assumption of the model is that participants could visually estimate the volumes of the different objects without noise. This could contribute to the discrepancy in s.d., because uncertainty in estimating object volume would be expected to increase uncertainty in the conditional probability of mass, exceeding the natural variation in mass of objects of that precise volume.”

2. Please include the details of the MLE procedure (line 238 onwards). Was it based on the predicted distribution of the perceived weight ratio at the level of individual trials?

That's correct. We have expanded this section in Methods:

“We used non-linear optimization to obtain maximum likelihood values for each participant of parameters β , c_m , and s , the slope, intercept and s.d. describing the prior distribution, and c_b and c_σ , the constants of proportionality for bias and s.d. respectively. Specifically, for each participant, we used a custom-coded pattern search algorithm to iteratively search for parameters that minimized the summed negative log likelihoods of the reported mass ratio estimates on each trial, based on the the mass and volumes of each pair of lifted items. To protect against local minima, each search was repeated with 100 sets of randomized starting parameter values, and the parameters corresponding to the global maximum likelihood selected. To enhance search efficiency, the fitting algorithm used the parameterization $\{\beta, c_m, s^2, \log_{10}(c_m), \log_{10}(c_\sigma)\}$ and the obtained maximum likelihood values were subsequently transformed to values of $\{\beta, c_m, s, c_m, c_\sigma\}$ for ease of interpretation. Parameters of the illustrative bivariate normal prior shown in Fig. 2 (contours) were medians calculated from maximum likelihood fitted parameter values with $\mu_v, \sigma_{mv} = 5$.”

3. Typo in line 91: “... the object of larger volume was on average estimated as heavier ...”. Should be “smaller volume”.

Fixed.

4. The modeling assumption here (especially for the material-weight illusion) would suggest that haptic encoding of weight/force is dynamically adjusted based on visual (size, material) cues. If so, one would expect weight discrimination threshold profile changes depending on visual contexts. This is outside the scope of this study, but I was wondering if the author could comment a bit more on this (e.g., if there's any empirical evidence along this line in the literature).

Thanks for this suggestion. Very few studies to our knowledge have reported weight discrimination performance in this context, but what there is seems consistent with our hypothesis. We have added the following to the discussion:

“The present study draws on previous work that has formulated the goal of efficient coding in terms of allocation of Fisher Information (Ganguli & Simonelli, 2014; Morais & Pillow, 2018; Wei & Stocker, 2015), which can in turn be related to discriminability via the Cramér-Rao bound (Series et al., 2009). The model of the size-weight illusion presented here therefore makes clear predictions for discrimination thresholds (JNDs), in addition to estimation bias and variability. However, relatively few empirical studies

have measured weight discrimination while varying secondary object properties such as size. One important finding (Ross & Gregory, 1970) is that weight discriminability is maximal for "non-illusory" objects, i.e. those whose apparent weight on lifting is the same whether visual evidence about the object's material and size is present or absent (Ross, 1969). This aligns quite precisely with predictions of the efficient coding model of the SWI and MWI, in which biases arise from and vary in proportion to the gradient of (squared) discriminability (Methods, Eq. 4), with the result that peak discriminability coincides with zero estimation bias. A further prediction of the model, that might be tested in future work, is that this peak corresponds to the maximum of the conditional prior density, i.e., that the "non-illusory" weight for an object is the weight that is most probable given its visual properties, based on natural statistics."

5. I found the additional models with factorized and joint efficient coding on weight and volume to be particularly interesting. As the author noted, there has been little prior research addressing this issue (multivariate code) empirically. However, here all three models make very similar predictions in terms of bias/variance prediction (Fig. S2). The author might consider highlighting this discussion more in the main text.

Thanks for this point. We have expanded the Discussion to highlight the alternative encodings and suggest an avenue for further investigation:

"The present results do not uniquely specify a coding scheme for the relevant object properties, and a number of different encodings consistent with the efficient coding goal are compatible with the observed biases and variability in estimation. In addition to the conditional density model of the SWI presented in the main text, which implicitly assumes that object volume is encoded separately from object mass, we also considered a variant that encodes the two properties jointly, by factorizing their joint prior density into two orthogonal components, as well as a variant where encoding of mass is efficient with respect to the joint, instead of conditional, prior. These alternative models were found to make qualitatively similar predictions consistent with the SWI, and fitting showed them to have similar compatibility with the data (see Supplementary Text for full details and model comparisons). While the predictions for bias differed only weakly within the experimental range of masses and volumes, larger deviations might be observed for more extreme pairings of object properties, potentially allowing future studies to better discriminate between alternative encoding schemes. However the largest discrepancies between models will tend to coincide with the lowest prior probabilities (e.g., objects with unnaturally high or low density), so could prove challenging to evaluate experimentally."